# Prevalence and associated risk factors of intestinal parasitic infections among children in pastoralist and agro-pastoralist communities in the Adadle woreda of the Somali Regional State of Ethiopia

Kayla C. Lanker[1,2☯], Abdifatah M. Muhummed[1,2,3☯], Guéladio Cissé[2,4], Jakob Zinsstag[1,2], Jan Hattendorf[1,2], Ramadan Budul Yusuf[3], Shamil Barsenga Hassen[3], Rea Tschopp[1,2,5], Pascale Vonaesch[6] *

1 Human and Animal Health Unit, Swiss Tropical and Public Health Institute, Basel, Switzerland, 2 Faculty of Science, University of Basel, Basel, Switzerland, 3 Jigjiga University One Health Initiative, Jigjiga University, Jigjiga, Ethiopia, 4 Ecosystem Health Sciences Unit, Swiss Tropical and Public Health Institute, Basel, Switzerland, 5 One Health Unit, Armauer Hansen Research Institute, Addis Ababa, Ethiopia, 6 Department of Fundamental Microbiology, University of Lausanne, Lausanne, Switzerland

☯ These authors contributed equally to this work.
* pascale.vonaesch@unil.ch

**Data Availability Statement:** All data can be found in the Supplementary Material of this article.

## Abstract

### Background

Intestinal parasitic infections (IPIs) can cause illness, morbidity, and occasional mortality in children. Agro-pastoralist and pastoralist children in the Somali Regional State of Ethiopia (ESRS) are especially at risk for IPIs, as access to safe water, sanitation, and health services is lacking. Minimal data on the prevalence of IPIs and associated risk factors exists in this region.

### Methodology

We assessed the prevalence of IPIs and associated risk factors during the wet season from May-June 2021 in 366 children aged 2 to 5 years in four agro-pastoralist and four pastoralist *kebeles* (wards) in Adadle *woreda* (district) of the Shebelle zone, ESRS. Household information, anthropometric measurements, and stool samples were obtained from included children. Parasites were identified microscopically using Kato-Katz and direct smear methods. Risk factors were assessed using general estimating equation models accounting for clustering.

### Principal findings

Overall prevalence of IPIs was 35%: 30.6% for single infections and 4.4% for poly-parasitic infections. Intestinal protozoan prevalence was 24.9%: 21.9% *Giardia intestinalis*, and 3.0% *Entamoeba* spp.. Intestinal helminth prevalence was 14.5%: 12.8% *Ascaris lumbricoides*, 1.4% hookworm (*Ancylostoma duodenale* /*Necator americanus.*), and 0.3% *Hymenolepis*

**Funding:** This study received financial support from the Swiss Agency for Development and Cooperation (SDC) (no. 7F-09057.01.02 to JZ), the Nutricia Research Foundation (no. 2019-20 to PV), the Forschungsfonds of the University of Basel (Fellowship in 2020 to PV), the SNSF Return Grant (no. P3P3PA_177877 to PV) and the Eccellenza Fellowship (no. PCEFP3_194545 to PV). The funders had no role in study design, data collection and analysis, decision to publish, or preparation of the manuscript.

**Competing interests:** The authors have declared that no competing interests exist.

*nana*. *G. intestinalis* infection was associated with drinking water sourced from the river (aOR 15.6, 95%CI 6.84, 35.4) and from collected rainwater (aOR 9.48, 95%CI 3.39, 26.5), with toilet sharing (aOR 2.93, 95%CI 1.36, 6.31) and with household ownership of cattle (1–5 cattle: aOR 1.65, 95%CI 1.13, 2.41; 6+ cattle: aOR 2.07, 95%CI 1.33, 3.21) and chickens (aOR 3.80, 95%CI 1.77, 8.17). *A. lumbricoides* infection was associated with children 36 to 47 months old (aOR 1.92, 95%CI 1.03, 3.58).

## Conclusions/significance

Improving access to safe water, sanitation, and hygiene services in Adadle and employing a One Health approach would likely improve the health of children living in (agro-) pastoralist communities in Adadle and the ESRS; however, further studies are required.

### Author summary

Intestinal parasitic infections remain a silent threat to the health and life-trajectories of children living in areas with inadequate access to clean water, proper sanitation, and hygiene facilities, including the Somali Regional State of Ethiopia. A large majority in this region live as pastoralists (semi-mobile animal herders), in close contact with their animals and nature, at risk for climate-related threats like drought and flooding, and at risk for infectious agents like intestinal parasites. We assessed the prevalence of intestinal parasitic infections in pastoralist children in the Adadle district of the Somali Regional State of Ethiopia (ESRS), and the individual and household-level factors associated with these infections. We found that locally collected water, shared toilets, along with ownership of cows and chickens increased the risk for having an intestinal parasitic infection with *Giardia intestinalis*, which can cause diarrhea and is transmitted through water, food, and soil that have been contaminated by the feces of infected humans and animals. If access to clean water, sanitation, and hygiene infrastructure is not improved, these infections remain recurrent in these communities and their animals, continually affecting the health of children. This study is one of few involving pastoralists in this region, hopefully lending guidance to regional public health policies.

## Introduction

Intestinal parasitic infections (IPIs) remain one of the most common infectious diseases in humans and animals globally with a high burden of disease and occasional mortality [1,2]. Children are especially at risk for IPIs due to changes in nutrition, age-specific behaviours around play and hygiene, and their developing immune systems [1]. Additional risk factors include lack of access to clean water, inadequate sanitation and hygiene, lack of knowledge on transmission pathways and lack of epidemiological surveillance [2]. While the morbidity of specific IPIs varies, children chronically infected can develop iron deficiency (anemia), vitamin A, and other micronutrient deficiencies, become undernourished, experience prolonged diarrheal episodes (dysentery), develop cognitive delays, become stunted, and may even die as a result of acute or chronic infection [1,3]. The majority of IPIs are transmitted via the faecal-oral route, either through faecally contaminated hands, food, water, vegetation, other surfaces, and aerosolized particles [2,4–6]. Many human intestinal parasites are also zoonotic and can

be transmitted to or from other animals, some of which can also cause illness and occasional death in the animal, and as well as economic loss for the animal's owner [7–10].

In Ethiopia, prevalence of IPIs has been found to be between 42% and 53% in children under five years of age [6,11–17] and school-aged children [18–23], from which associated risk factors were unsafe or contaminated drinking water [6,13], no sanitation facilities and open defecation [6,11,12], poor hygiene and knowledge on transmission [6,11,13], living in crowded homes [12], eating raw vegetables [6,12,17], and coming into contact with contaminated soils [6,17]. However, a 2019 systematic review found that in several regions of Ethiopia, including the Ethiopian Somali Regional State (ESRS), no prevalence estimates of IPIs in children were published or available [24]. Indeed, to date, there is a single study assessing undernutrition and prevalence of IPIs in pastoralist children in the ESRS during the 2019 drought season [25]. There is no data on prevalence in the wet season nor information on associated risk factors of IPIs, leaving a knowledge gap regarding IPIs in pastoralist communities in the ESRS.

Pastoralist and agro-pastoralist communities make up the majority of the total population of the ESRS [26,27]. Communities are tight-knit and often blood related, often sharing child-care, water and food resources, and medicines. They rely mainly on their animals for food (milk and meat) and livelihood, are semi-mobile and lack access to safe water and sanitation as well as health services [25,28]. As pastoralists live in close connection and proximity to one another, their animals, and their environment year-round, it is expected that the sharing of microbial species, including intestinal parasites, is high. This study assessed the prevalence of intestinal parasitic infections and associated risk factors in children aged 2 to 5 years in pastoralist and agro-pastoralist communities in the Adadle *woreda* (district) of the Shabelle zone of the Ethiopian Somali Regional State (ESRS). This study is part of the Jigjiga One Health Initiative to improve the health of humans, animals, and their environments among pastoralist communities in the ESRS, with a specific focus on improving the health of children in the ESRS.

## Methods

### Ethics statement

Ethics approval for this study was obtained from the Review Committee of the University of Jigjiga in Ethiopia (JJU-RERC030/2020) and the Review Committee of Armauer Hansen Research Institute in Addis Ababa, Ethiopia (AF-10-015), and from the Swiss Ethics Committee of Northwest and Central Switzerland (Ethikkommision Nordwest- und Zentralschweiz; REQ-2020-00608). Oral or written consent was obtained from the parent of all participating children before study enrolment. If a child was found to have an intestinal parasitic infection, the child's mother was given anti-parasitic medications (mebendazole, albendazole or metronidazole) to administer to her child. Data was recorded on the Open Data Kit and stored on a secured server at the Swiss TPH in Basel. All identifying information was kept with the local study team in Ethiopia and is securely stored in a closed cupboard.

### Study design

**Study area.** The study was a cross-sectional study in the Adadle *woreda* (district) in the Shabelle zone of the Somali Regional State of Ethiopia and was carried out from May 2021 until June 2021 during the wet season, known locally as *gu-'ga*. Adadle is located 17 km from the city of Gode in the lowlands of the Wabi Shabelle River subbasin and experiences a mean annual temperature of 32˚C and a mean annual rainfall of 300 mm. The altitude is 300–500 meters above sea-level and 80% of the land is flat, while 20% is undulated. The pastoralist and agro-pastoralist communities in ESRS and Adadle rely mainly on their animals for food and livelihood and live largely outside of modern systems.

**Sample size.** The sample size was determined based on several components, as this study was part of a larger project studying antimicrobial resistance and the microbiome from a One Health perspective. A previous study in children aged 2 to 5 years in this region showed a prevalence of intestinal parasitic infections of 42% [25]. As we expected clustering on the *kebele* (ward) level, the intra-cluster correlation coefficient was assumed to be 0.15. We chose a 95% confidence interval, power of 80%, alpha of 0.05, and a margin of error of 10%. Based on these factors and to achieve the necessary sample size for all aims of the larger research project, we calculated a sample size of 360 eligible children, 180 each in pastoralist and agro-pastoralist communities.

**Selection criteria.** Stakeholders (community leaders and health officials) in Adadle were approached at each level (district, ward, sub-ward) and invited to help determine which kebeles in Adadle could be included in the present study. From the 15 kebeles in Adadle, four pastoralist (Malkasalah, Todob, Harsug, Kulmis) and four agro-pastoralist (Bursaredo, Gabal, Higlo, Boholhagare) kebeles were randomly selected (**Fig 1**). A pre-enrolment screening was carried out at the local health centre in each kebele, for all children aged 2 to 5 years. Medication and hygienic supplies were given as incentive to each household to attend the pre-screening, regardless of further participation in the study. All children were screened for stunting (height for age) and wasting (weight for height), according to the 2006 WHO growth standards [29]. All screened children who were severely wasted based on their weight for height z-score (WHZ < -3) or severely stunted based on their height for age z-score (HAZ < -3) were automatically included in the study. Screened children not presenting as severely stunted and/or severely wasted were randomly selected (random number generation) from an excel file containing all pre-screened households/children in the community, until the maximum sample

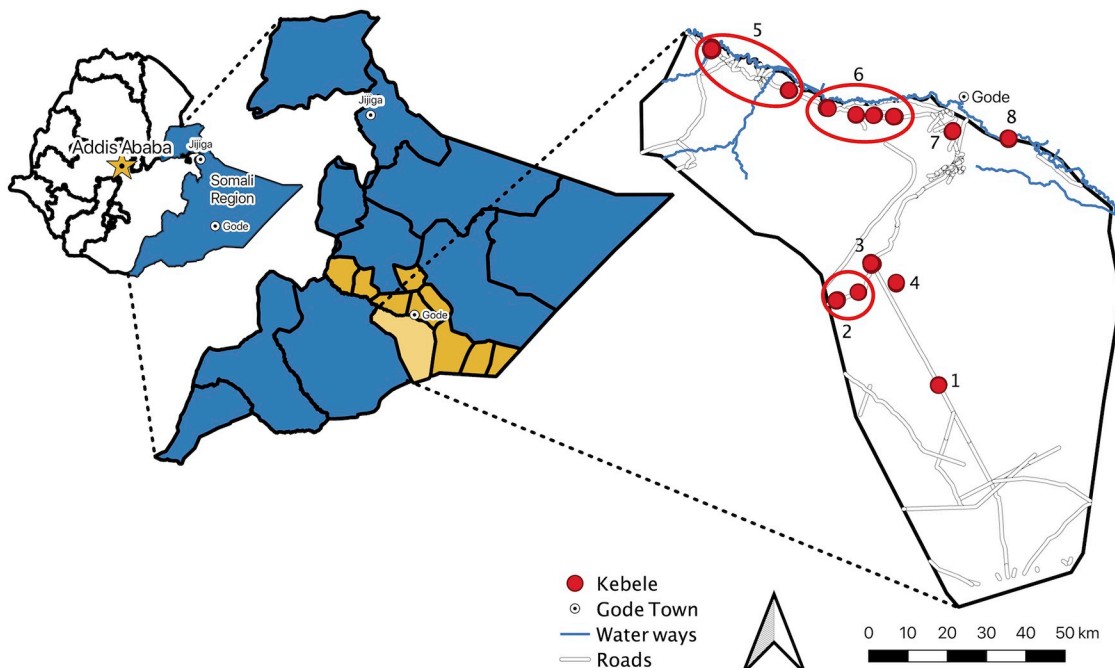

**Fig 1. Map showing included kebeles in Adadle woreda in the Shabelle zone of the Somali Regional State, Ethiopia.** Kebele locations are based on geopoints collected during the study. Kebeles 1–4 are pastoralist communities (1: Todob; 2: Kulmis; 3: Malkasalah; 4: Harsug) and kebeles 5–8 are agro-pastoralist communities (5: Gabal; 6: Bursaredo; 7: Higlo; 8: Boholhagare). All public geographic data files (borders, cities, roads, and waterways) were downloaded from the open source Humanitarian Data Exchange (HDX) [39]. Map created in QGIS3 [38].

size was reached for each kebele and the overall study. Children were excluded if they were older than 5 years, younger than 2 years and/or had taken antibiotics in the last 14 days.

## Anthropometric measurements

Anthropometric measurements of height, weight, and mid-upper arm circumference (MUAC) were measured at least twice, or until the measurements were within 1 cm, 100 g, or 1 mm of one another, respectively. The measurements were then averaged for each child. Height was measured by having the child stand up straight against a WHO-standard wooden measuring board [29] set on flat ground. Attention was given that the child was looking forward, had relaxed shoulders, straight legs, no shoes on, and arms at their side. Weight was measured by having the child stand alone in the middle of a WHO-standard scale [29] without shoes, and with light clothing. If the child was not willing or able to be weighed alone, then the mother was first weighed alone and then weighed again with the child in her arms. The final weight of the child was therefore the weight of the mother alone subtracted from the weight of mother and child together. The scale was calibrated daily against known weights to ensure accuracy on all study days. The mid-upper arm circumference (MUAC) was measured using a WHO-standard tape measurer [29]. In the pre-screening, z-scores for height-for-age (HAZ; stunting) and weight-for-height (WHZ; wasting) were calculated using the 2006 WHO growth standard tables [29]. Following the end of data collection, HAZ, WAZ, and WHZ were recalculated using the R package 'zscorer' [30]. Stunting, wasting, and underweight were defined as HAZ, WHZ, WAZ $< -2$, while acute malnutrition (low MUAC) was defined as MUAC $< 12.5$ cm.

## Data collection

A detailed questionnaire using the Open Data Kit (ODK) software [31] was administered in the local language by field workers to the mother of the child. The questionnaire was divided into several sections: (1) child anthropometric measures, current health, and health history of the child, (2) household characteristics and assets, and the type and number of household animals, (3) WASH behaviours of the child and household, (4) a nutritional survey of the child, and finally (5) breastfeeding and birth history of the mother and child. Birth records often do not exist in pastoralist communities; therefore, age in months was estimated based on group discussions with the child's mother, other family, and community members to determine seasonal (floods, month of year, drought) and festive events (Ramadan, other religious events) that occurred before or after the birth of the child. Geo-reference points for spatial visualizations were collected for most households.

## Stool sample collection and parasitological analyses

Stool samples were collected by the mothers in sterile specimen cups after detailed instruction by a trained field worker in the local language. The samples were aliquoted and then immediately transferred to a cold box containing ice. All parasitology analyses were performed in the field in each kebele health clinic by trained parasitologists from Jigjiga University. The aliquot for parasitology was divided into four portions. One portion was used for the quantification of helminth ova using the thick-smear Kato-Katz detection method [32] under a microscope at 10x and 40x magnification, which allows for the detection of common helminth species such as hookworm (*Ancylostoma duodenale*/*Necator americanus*), *Hymenolepis nana*, *Trichuris trichiura*, *Ascaris lumbricoides*, and *Schistosoma mansoni*. Another two portions were used for the detection of protozoan cysts and trophozoites, such as *Giardia intestinalis* and *Entamoeba* spp. using the direct smear method in duplicate. Briefly, a drop of normal saline and a drop of Lugol's iodine were each placed on one half of a single sterile glass slide. Then, using a cotton

swab, a small amount of stool sample was smeared onto each drop, covered with a glass slip, and inspected under a microscope at 10x magnification. Although microscopy can only be used to identify *Giardia* at the genus level, *Giardia intestinalis* is the only species of *Giardia* that has been found to infect humans. Therefore, all *Giardia* identified through microscopy in the human faecal samples are referred to as *Giardia intestinalis* throughout the manuscript. The final portion of stool was fixed and stored in SAF (sodium acetate–acetic acid-formalin) at 4˚C as a backup [33]. The consistency of each stool sample was determined by a trained parasitologist.

## Statistical analysis

All statistical analyses were performed in R Statistical software version 4.0.4 [34]. Tables were generated using the gtsummary package [35]. Briefly, the ODK questionnaire data (including anthropometric data) and parasitology data were merged according to each child's identification number. We used the available case population for all analyses. For household-level metrics, missing data was addressed by applying the completed questionnaires of siblings in the same household. Missing data for individual metrics (such as anthropometric metrics) were not able to be recovered and these observations were therefore left out of analyses. Individual observations where the mother answered "don't know" are listed as "Unknown" in descriptive tables. Categorization of numerical variables, such as ownership of cattle, were based on the median value as a cut-off point. Univariable and multivariable analysis were carried out using the general estimating equation logistic regression model for binary outcomes using the geepack package [36], to account for clustering on the kebele level. Variables in the multivariable model were pre-selected based on the published literature and complemented with variables associated in the univariable analysis (p < 0.2) [37]. Spatial visualizations were performed on QGIS3 [38] and public geographic data (borders, cities, roads, and waterways) was downloaded from open source Humanitarian Data Exchange (HDX) [39].

## Results

A total of 366 children aged between two to five years from 270 households spread between eight kebeles in the Adadle woreda (district), Shabelle zone of the Ethiopian Somali Regional State (ESRS) were included in the present study, of which 184 children were from pastoralist and 182 from agro-pastoralist communities. During the screening process, 7 children presented as severely stunted, 27 as severely wasted, and 3 as severely stunted and severely wasted, all of whom were included in the study. For all included children, parasitology reports were documented. However, for twenty-one children, a questionnaire was not recovered or completed, due to logistical issues with the software and field activities. Therefore, 345 children completed both a questionnaire and parasitology report (**S1 Fig**). By applying household-level data from completed questionnaires of siblings in the same household, 13 of the 21 children with a missing questionnaire were able to be recovered for the household-level descriptive analyses (N = 358; **S1 Fig**).

### Participant characteristics

Participant characteristics are summarized in **Table 1**. The age and sex of enrolled children were distributed evenly, including between pastoralist and agro-pastoralist communities. Only 10.9% of children had a complete vaccination at the time of sampling. Only 8.4% of children were exclusively breastfed (EBF) for the first 6 months. The median duration of breastfeeding was 12 months (Range: 2–28 months). Supplementary milk from household animals (cows, goats, camels) was given in the first 6 months of life to 57.5% of children, while 21.7% received

**Table 1. Characteristics of (agro-) pastoralist children aged 2 to 5 years living in Adadle woreda, Shabelle zone, Somali Regional State, Ethiopia.**

| Characteristic | N = 345 |
|---|---|
| **Age** | |
| 23–35 months | 120 (34.8%) |
| 36–47 months | 106 (30.7%) |
| 48–60 months | 119 (34.5%) |
| **Sex** | |
| Female | 169 (49.0%) |
| Male | 176 (51.0%) |
| **Vaccination status** | |
| Complete vaccination | 37 (10.9%) |
| Incomplete vaccination | 301 (89.1%) |
| Unknown | 7 |
| **Exclusively breastfed for 6 months** | 29 (8.4%) |
| **Complementary milk starting when** | |
| Before 6 months | 196 (57.5%) |
| After 6 months | 74 (21.7%) |
| Never gave other milk | 71 (20.8%) |
| Unknown | 4 |
| **Complementary food starting when** | |
| Before 6 months | 80 (23.9%) |
| After 6 months | 255 (76.1%) |
| Unknown | 10 |

EBF, Exclusive breastfeeding; Complementary food included mostly soft staple foods cooked in milk. Overall statistics are shown, as group differences were minimal. Data collected in the wet season 2021.

Data are presented as n (%).

supplementary milk after 6 months, and 20.8% were never given supplementary milk prior to weaning. Supplementary animal milk was given as early as 1 month and as late as 28 months, with the median start time at 1 month old. Supplementary foods (mostly soft staple foods like porridge, rice, potatoes, and injera) were started in 23.9% of children in the first 6 months of life, while the rest (76.1%) received supplementary food after 6 months of age. Supplementary food was given as early as 1 month but as late as 40 months, with a median start time at 6 months.

Height, weight, and mid-upper arm circumference (MUAC) were similarly distributed between pastoralist and agro-pastoralist children (**Table 2**). Of the included children with anthropometric metrics (N = 345), 10 (3%) were severely stunted (HAZ < -3) and 30 (9%) were severely wasted (WHZ < -3), of which 3 children were both severely stunted and wasted. The overall stunting (HAZ < -2), wasting (WHZ < -2), underweight (WAZ < -2) and low MUAC (MUAC <12.5 cm) were 14%, 30%, 17%, and 2.3%, respectively. Only 5% (17/345) of children were both wasted and stunted.

## Household and Kebele characteristics

The included agro-pastoralist kebeles all reside along or near the Shebelle River (**Fig 1**); two kebeles (7: Higlo; 8: Boholhagare) lived along the Shebelle River and were approximately 16 km downstream from Gode town, one kebele (6: Bursaredo) was near an agricultural zone 20–40 km from Gode banking the Shebelle River, and the last kebele (5: Gabal) banked the

**Table 2. Anthropometric characteristics of agro-pastoralist and pastoralist children aged 2 to 5 years living in Adadle woreda, Shabelle zone, Somali Regional State, Ethiopia.**

| Characteristic | Agro-pastoralist N = 168 | Pastoralist N = 177 | Overall N = 345 |
|---|---|---|---|
| Height (cm) | 96 (90, 102) | 98 (90, 105) | 97 (90, 104) |
| Weight (kg) | 12.60 (11.38, 13.90) | 13.30 (11.60, 15.00) | 12.80 (11.40, 14.50) |
| HAZ | -0.37 (-1.45, 0.41) | 0.02 (-0.97, 0.78) | -0.18 (-1.21, 0.59) |
| Stunted | 26 (15%) | 21 (12%) | 47 (14%) |
| WHZ | -1.55 (-2.18, -0.98) | -1.33 (-2.03, -0.65) | -1.41 (-2.15, -0.82) |
| Wasted | 53 (32%) | 49 (28%) | 102 (30%) |
| WAZ | -1.34 (-1.95, -0.72) | -0.83 (-1.62, -0.22) | -1.06 (-1.78, -0.40) |
| Underweight | 39 (23%) | 19 (11%) | 58 (17%) |
| MUAC (cm) | 14.00 (13.50, 14.72) | 14.20 (13.60, 14.90) | 14.10 (13.50, 14.90) |
| Low MUAC | 5 (3.0%) | 3 (1.7%) | 8 (2.3%) |

HAZ, height for age z-score; Stunted: HAZ < -2 SD; WHZ, weight for height z-score; Wasted: WHZ < -2 SD; WAZ, weight for age z-score; Underweight: WAZ < -2 SD; MUAC, mid-upper arm circumference; Low MUAC: MUAC <12.5 cm.

Data are presented as median (IQR) or n (%) unless otherwise stated.

Shebelle River 50–70 km upstream of Gode town. The included pastoralist kebeles (1: Todob; 2: Kulmis; 3: Malkasalah; 4: Harsug) did not reside near any rivers, but did flank road systems, and were between 48–75 km from Gode town (**Fig 1**). Households (regardless of if pastoralist or agro-pastoralist) spoke exclusively the Somali language (100%), were of the Muslim faith (100%), were majority illiterate (head of household, 74.3%; mother of child, 76.0%), and a majority owned a mobile phone (62.3%) (**S1 Table**).

Household water, sanitation, and hygiene (WASH) characteristics showed some differences between pastoralist and agro-pastoralist communities and are summarized in **Table 3**. The majority (85.3%) of agro-pastoralist communities sourced their water from the Shebelle River, while the majority (89.0%) of pastoralist households sourced their water from rain collected in open shallow wells or birkads. Only a small minority sourced their water from a borehole, tank truck or natural spring (agro-pastoralist: 5.6%, pastoralist: 9.4%). More agro-pastoralist (46.3%) than pastoralist (14.9%) treated their water, through chlorination. Agro-pastoralists almost exclusively practiced open defecation (99.4%), while pastoralists practiced open defecation 92.8% of the time and 7.2% indicated they used a type of pit latrine. Toilet sharing, meaning the sharing of a latrine or defecation spot, was shared by a minority in both agro-pastoralist (12.4%) and pastoralist (5.5%) communities. Waste disposal was similar between agro-pastoralists and pastoralists; the majority dumped their waste (84.1%), either in their compound, river or in the open, while few burned their waste (15.9%). Household ownership of soap was also quite low, with 56.0% indicating they do not usually have soap in the house. Finally, hygienic characteristics of the households were similar, with most mothers reporting that they washed their child's hands with water only (92.3%), and few with water and soap (7.7%).

Agro-pastoralists and pastoralists in Adadle and the ESRS depend greatly on livestock animals as a means of food and livelihood. Household ownership of domestic animals, which includes in the present study cattle, camel, goats, sheep, donkeys, and chickens, varied in quantity between the agro-pastoralist and pastoralist communities in Adadle (**Fig 2** and **S2 Table**). Agro-pastoralists owned more cattle, sheep, donkeys, and chickens, while pastoralists owned more goats and camels and no chickens. Livestock animal ownership ranged from zero to 149 animals, with a median of 25 animals, and more animals were kept inside the house or compound of pastoralist households (78.5%) than agro-pastoralist households (57.6%) (**S2 Table**).

**Table 3. Household WASH characteristics of agro-pastoralist and pastoralist children aged 2 to 5 years living in Adadle woreda, Somali Regional State, Ethiopia.**

|  | Characteristic | Agro-pastoralist N = 177 | Pastoralist N = 181 | Overall N = 358 |
|---|---|---|---|---|
| WATER | **Source of drinking water** |  |  |  |
|  | Borehole/Spring/Tank truck | 10 (5.6%) | 17 (9.4%) | 27 (7.5%) |
|  | Rainwater in Birkads | 16 (9.0%) | 161 (89.0%) | 177 (49.4%) |
|  | River water | 151 (85.3%) | 3 (1.7%) | 154 (43.0%) |
|  | **Treatment of water (Yes)** | 82 (46.3%) | 27 (14.9%) | 109 (30.4%) |
| SANITATION | **Toilet type** |  |  |  |
|  | Pit latrine | 1 (0.6%) | 13 (7.2%) | 14 (3.9%) |
|  | Outdoor | 176 (99.4%) | 168 (92.8%) | 344 (96.1%) |
|  | **Shared toilet (Yes)** | 22 (12.4%) | 10 (5.5%) | 32 (8.9%) |
|  | **Waste disposal** |  |  |  |
|  | Burned | 29 (16.4%) | 28 (15.5%) | 57 (15.9%) |
|  | Dumped | 148 (83.6%) | 153 (84.5%) | 301 (84.1%) |
| HYGIENE | **Household owns soap** |  |  |  |
|  | Yes | 83 (47.2%) | 74 (40.9%) | 157 (44.0%) |
|  | No | 93 (52.8%) | 107 (59.1%) | 200 (56.0%) |
|  | Unknown | 1 | 0 | 1 |
|  | **Child hand washing method** |  |  |  |
|  | With water | 151 (88.3%) | 172 (96.1%) | 323 (92.3%) |
|  | With water and soap | 20 (11.7%) | 7 (3.9%) | 27 (7.7%) |
|  | Unknown | 6 | 2 | 8 |

Birkads: open shallow wells or ditches; Treatment of water was mostly through chlorination.

Data are presented as n (%).

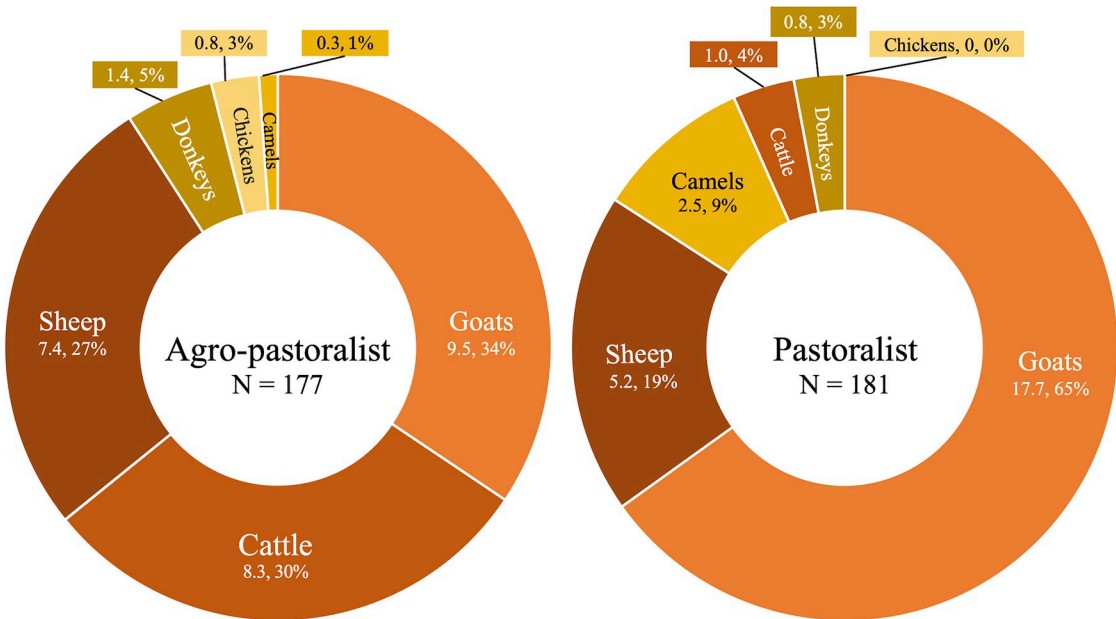

**Fig 2. Average household animal herd makeup of agro-pastoralist and pastoralist children aged 2 to 5 years living in Adadle woreda, Somali Regional State, Ethiopia.** Numbers are the mean and mean percentage of each type of animal owned per participant per group (pastoralist or agro-pastoralist). N = 358.

**Table 4. Prevalence of intestinal parasitic infections during the 2021 wet season in agro-pastoralist and pastoralist children aged 2 to 5 years living in Adadle woreda, Somali Regional State, Ethiopia.**

| Characteristic | Agro-pastoralist N = 182 | Pastoralist N = 184 | Overall N = 366 |
|---|---|---|---|
| **Overall** | | | |
| Parasite free | 115 (63.2%) | 123 (66.8%) | 238 (65.0%) |
| Single parasite | 57 (31.3%) | 55 (29.9%) | 112 (30.6%) |
| Poly-parasite | 10 (5.5%) | 6 (3.3%) | 16 (4.4%) |
| **Protozoa** | | | |
| *Giardia* spp. | 44 (24.2%) | 36 (19.6%) | 80 (21.9%) |
| *Entamoeba* spp. | 3 (1.6%) | 8 (4.3%) | 11 (3.0%) |
| **Helminths** | | | |
| *Ascaris lumbricoides* | 26 (14.3%) | 21 (11.4%) | 47 (12.8%) |
| Hookworm (*Ancylostoma duodenale /Necator americanus*) | 4 (2.2%) | 1 (0.5%) | 5 (1.4%) |
| *Hymenolepis nana* | 0 (0.0%) | 1 (0.5%) | 1 (0.3%) |

Data are presented as n (%).

### Intestinal parasitic infection prevalence

The overall prevalence of intestinal parasitic infections (IPIs) was 35%, with a prevalence of 30.6% for single parasitic infections and 4.4% for poly-parasitic infections (coinfection with protozoan and helminth species). The infection pattern is summarized in **Table 4**. Intestinal protozoan infections were detected in 24.9% of children, while intestinal helminth infections were detected in 14.5% of children. The prevalence of protozoan infections was 21.9% *Giardia intestinalis* and 3.0% *Entamoeba* spp.. Of the *G. intestinalis* infections detected, 65% were in the cystic stage and 35% in the trophozoite stage. *G. intestinalis* prevalence was slightly higher in agro-pastoralist communities (24.2%) compared with pastoralist communities (19.6%), although this difference was not significant in univariate or multivariate analyses (**Table 5**). The overall prevalence of helminth infections was 12.8% *Ascaris lumbricoides*, 1.4% hookworm (*Ancylostoma duodenale /Necator americanus*), and 0.3% *Hymenolepis nana*. Both *A. lumbricoides* and hookworm (*A. duodenale /N. americanus*) had a slightly higher prevalence in agro-pastoralist communities compared with pastoralist communities, although this difference was not significant in univariate or multivariate analyses (**Table 6**). Poly-parasitic infections (4.4%) were all found to be a co-infection of a protozoan with a helminth species, of which 25% (4/16) were *Entamoeba* spp. with the helminth *A. lumbricoides* and 75% (12/16) *G. intestinalis* with a helminth (*A. lumbricoides* 9/16; *H. nana* 1/16; hookworm 2/16).

### Factors associated with *Giardia intestinalis* infection

In univariate analysis, *G. intestinalis* infection was associated with children aged 36 to 47 months, source of drinking water, ownership of cattle and ownership of chickens. In multivariate analysis, *G. intestinalis* infection was associated with water source, toilet sharing, ownership of cattle and ownership of chickens (**Table 5**). Sourcing drinking water from rainwater stored in birkads (open shallow wells/ditches) had an aOR of 9.48 [3.39, 26.5], p-value < 0.001, while sourcing from river water had an aOR of 15.6 [6.84, 35.4], p-value < 0.001, when compared to sourcing water from a spring, tank truck, or borehole. Toilet sharing had an aOR of 2.93 [1.36,6.31], p-value = 0.006. Household ownership of cattle revealed an aOR of 1.65 [1.13 2.41], p-value = 0.009, when owning 1–5 cattle, and an aOR of 2.07 [1.33, 3.21], p-value = 0.001, when owning more than 6 cattle, compared with owning no cattle. Similarly, owning chickens had an aOR of 3.8 [1.77, 8.17], p-value < 0.001. All further

**Table 5. Univariate and multivariate analysis of *Giardia intestinalis* in agro-pastoralist and pastoralist children aged 2 to 5 years living in Adadle woreda, Somali Regional State, Ethiopia.**

| Characteristic | n/N (% Positive) | Univariate | | | Multivariate | | |
|---|---|---|---|---|---|---|---|
| | | OR | 95% CI | p-value | aOR | 95% CI | p-value |
| **Group** | | | | | | | |
| Agro-pastoralist | 41 / 168 (24) | — | — | | — | — | |
| Pastoralist | 35 / 177 (20) | 0.76 | 0.36, 1.63 | 0.49 | 2.52 | 0.73, 8.73 | 0.14 |
| **Sex** | | | | | | | |
| Female | 38 / 169 (22) | — | — | | — | — | |
| Male | 38 / 176 (22) | 0.95 | 0.62, 1.45 | 0.81 | 0.99 | 0.63, 1.56 | 0.97 |
| **Age** | | | | | | | |
| 23–35 months | 21 / 120 (18) | — | — | | — | — | |
| 36–47 months | 30 / 106 (28) | 1.86 | 1.08, 3.21 | **0.025** | 1.68 | 0.90, 3.16 | 0.11 |
| 48–60 months | 25 / 119 (21) | 1.25 | 0.81, 1.94 | 0.31 | 1.19 | 0.75, 1.91 | 0.46 |
| **Source of drinking water** | | | | | | | |
| Borehole/Spring/Tanktruck | 1 / 27 (3.7) | — | — | | — | — | |
| Rain water | 36 / 171 (21) | 6.93 | 2.41, 19.9 | <**0.001** | 9.48 | 3.39, 26.5 | <**0.001** |
| River water | 39 / 147 (27) | 9.39 | 2.64, 33.4 | <**0.001** | 15.6 | 6.84, 35.4 | <**0.001** |
| **Shared toilet** | | | | | | | |
| No | 66 / 315 (21) | — | — | | — | — | |
| Yes | 10 / 30 (33) | 1.89 | 0.94, 3.79 | 0.074 | 2.93 | 1.36, 6.31 | **0.006** |
| **Number of cattle** | | | | | | | |
| No | 23 / 130 (18) | — | — | | — | — | |
| 1–5 cattle | 29 / 128 (23) | 1.36 | 0.74, 2.52 | 0.32 | 1.65 | 1.13, 2.41 | **0.009** |
| 6+ cattle | 24 / 87 (28) | 1.77 | 1.12, 2.80 | **0.014** | 2.07 | 1.33, 3.21 | **0.001** |
| **Owns chickens** | | | | | | | |
| No | 67 / 327 (20) | — | — | | — | — | |
| Yes | 9 / 18 (50) | 3.88 | 2.63, 5.73 | <**0.001** | 3.80 | 1.77, 8.17 | <**0.001** |

OR = Odds Ratio; aOR = adjusted Odds Ratio; CI = confidence interval; Univariate and multivariate analyses performed using a general estimating equation logistic regression model for binary outcomes. N = 345.

variables included in multivariate analysis were non-significant. Univariate and multivariate analyses were not possible for *Entamoeba* spp., due to low positive numbers.

### Factors associated with *Ascaris lumbricoides* infection

In univariate analysis, *Ascaris lumbricoides* was associated with children aged 36 to 47 months and with owning more than 6 cattle (**Table 6**). In multivariate analysis, *A. lumbricoides* infection remained associated with children aged 36 to 47 months (aOR 1.92 [1.03, 3.58], p-value = 0.040) compared to children aged 23 to 35 months. All further variables included in multivariate analysis were non-significant. Univariate and multivariate analyses were not possible for the other helminth parasites due to low positive numbers.

### Discussion

To our knowledge, this is the first study assessing the impact of livestock animal keeping and WASH characteristics on the risk of intestinal parasitic infections (IPIs) in children 2 to 5 years of age living in pastoralist and agro-pastoralist communities in the ESRS. In addition, this study adds to the limited body of research on the health of pastoralist communities living

**Table 6. Univariate and multivariate analysis of *Ascaris lumbricoides* in agro-pastoralist and pastoralist children aged 2 to 5 years living in Adadle woreda, Somali Regional State, Ethiopia.**

| Characteristic | n/N (% Positive) | Univariate | | | Multivariate | | |
|---|---|---|---|---|---|---|---|
| | | OR | 95% CI | p-value | aOR | 95% CI | p-value |
| **Group** | | | | | | | |
| Agro-pastoralist | 26 / 168 (15) | — | — | | — | — | |
| Pastoralist | 20 / 177 (11) | 0.70 | 0.35, 1.39 | 0.30 | 2.03 | 0.62, 6.71 | 0.24 |
| **Sex** | | | | | | | |
| Female | 21 / 169 (12) | — | — | | — | — | |
| Male | 25 / 176 (14) | 1.17 | 0.69, 1.97 | 0.56 | 1.31 | 0.73, 2.38 | 0.37 |
| **Age** | | | | | | | |
| 23–35 months | 14 / 120 (12) | — | — | | — | — | |
| 36–47 months | 22 / 106 (21) | 1.98 | 1.12, 3.53 | **0.020** | 1.92 | 1.03, 3.58 | **0.040** |
| 48–60 months | 10 / 119 (8.4) | 0.69 | 0.23, 2.09 | 0.52 | 0.68 | 0.23, 2.01 | 0.48 |
| **Source of drinking water** | | | | | | | |
| Borehole/Spring/Tanktruck | 4 / 27 (15) | — | — | | — | — | |
| Rain water | 18 / 171 (11) | 0.68 | 0.24, 1.87 | 0.45 | 0.78 | 0.24, 2.52 | 0.67 |
| River water | 24 / 147 (16) | 1.12 | 0.36, 3.52 | 0.84 | 1.39 | 0.38, 5.01 | 0.62 |
| **Shared toilet** | | | | | | | |
| No | 39 / 315 (12) | — | — | | — | — | |
| Yes | 7 / 30 (23) | 2.15 | 0.64, 7.30 | 0.22 | 2.31 | 0.68, 7.88 | 0.18 |
| **Number of cattle** | | | | | | | |
| No | 13 / 130 (10) | — | — | | — | — | |
| 1–5 cattle | 16 / 128 (12) | 1.29 | 0.68, 2.44 | 0.44 | 1.19 | 0.58, 2.46 | 0.64 |
| 6+ cattle | 17 / 87 (20) | 2.19 | 1.01, 4.71 | **0.046** | 1.85 | 0.76, 4.51 | 0.18 |
| **Owns chickens** | | | | | | | |
| No | 41 / 327 (13) | — | — | | — | — | |
| Yes | 5 / 18 (28) | 2.68 | 0.77, 9.39 | 0.12 | 2.98 | 0.58, 15.2 | 0.19 |

OR = Odds Ratio; aOR = adjusted Odds Ratio; CI = confidence interval; Univariate and multivariate analyses performed using a general estimating equation logistic regression model for binary outcomes. N = 345.

in the ESRS and in the horn of Africa, the latter geographic zone encompassing an estimated 20–30 million humans living a pastoral lifestyle [26,27].

A recent study performed in the same district with pastoralist communities during the dry (drought) season showed slightly higher overall prevalence of IPIs (47%) than in our study (35%), although the prevalence of *G. intestinalis* and *A. lumbricoides* were comparable [25]. This difference is likely due to the effect of seasonality on transmission and prevalence of IPIs [40,41] as well as on nutritional status [42,43], which is known to have synergistic interactions with IPIs. In other regions of Ethiopia, prevalences of IPIs in pre-school aged children varied (between 17% and 58%), with most studies conducted in non-pastoralist communities, at hospitals or schools, and in wet or mountainous zones [6,11–17]. Pastoralists in Chad were found to have a higher prevalence of IPIs (60%) [44]. Ecological and cultural differences, such as behaviours around hygiene, animals, and food, as well as child rearing, weaning and outdoor play, may all impact the prevalence of IPIs in each study setting, making comparisons difficult.

Among the intestinal parasites detected in this study, *Giardia intestinalis* infection was the highest and was found to be associated with source of drinking water, toilet sharing, ownership of cattle, and ownership of chickens. Sourcing from river water or from rainwater collected in shallow open wells or ditches (birkhads) both showed a higher odds of *G. intestinalis* infection

compared to sourcing from a borehole, spring, or tank-truck, indicating a potential contamination of the water sources and/or where they are stored. Both water sources (river or rain) represent much higher chances of environmental and animal/human contamination, especially during rainy (flood) seasons, when soils contaminated with human and animal feces, due to high levels of open defecation, are carried into nearby rivers, streams, and open wells/ditches. Other studies have demonstrated that poor water quality or feces contaminated water contributes to *G. intestinalis* (and other parasites) transmission and prevalence in humans and animals [6–9,40]. In addition, water security in the region is a recurrent problem, with drought-intensity increasing every year due to climate change [45,46]. Therefore, in combination with drought-related water shortages and contamination of crucial and limited water sources, the water security situation in this region for pastoralists may worsen and impact human and animal health even more [47]. Our results therefore represent a strong indication for priority to be given to constructing and maintaining improved water sources for agro-pastoralists and pastoralists in Adadle woreda and the wider ESRS to reduce the transmission of *G. intestinalis* and other water-borne pathogens.

Household sharing of the toilet area was found to be associated with *Giardia intestinalis* infection, indicating that this may be a potential transmission point between households. Indeed, studies in Ethiopia, the UAE, and India all found sharing of the toilet area with multiple people or between households increased the individual risk for IPIs [48–50]. As *G. intestinalis* is transmitted via the faecal-oral route, behaviours surrounding defecation, such as style of latrine, maintenance of latrine, disposal of human waste, personal methods used to clean oneself, and handwashing behaviours, are all important factors that may contribute to transmission in shared toilets [48–50]. Paired with our finding that many households (56%) did not own soap, most mothers (92%) washed their child's hands with only water, and most households (84%) dumped their waste in either the compound, the river, or the open, appropriate sanitation and hygiene measures should be developed in cooperation with (agro-) pastoralist communities in Adadle, to reduce transmission of *G. intestinalis*.

Ownership of cattle was also found to be significantly associated with *Giardia intestinalis* infection. With increasing numbers of cattle, categorically evaluated, the odds of infection also increased. Similarly, ownership of chickens was significantly associated with *G. intestinalis* infection, although this result is limited by the small number of participants living in households owning chickens (N = 18) and the fact that no pastoralist participants owned chickens. The association of *G. intestinalis* with chicken ownership is therefore more applicable to agro-pastoralist communities. Our results are supported by the fact that *G. intestinalis* is a zoonotic parasite, with the ability to infect and be transmitted to and from many mammals, including in cattle, chickens, and other livestock [7,51–54]. Indeed, a recent study found frequent cases of animal syndromes in pastoralist livestock in Adadle woreda during a seven month study period, with cattle and sheep showing high rates of gastrointestinal diseases (42.8%) [28,55]. Transmission of *G. intestinalis* from cows or chickens to children could therefore occur through several indirect pathways: children playing around household animals may become exposed to animal feces containing parasitic cysts, eggs, or larva; animals and humans defecating in similar spots, including near shared water sources, may allow for cross-over contamination; and if proper hygiene measures (handwashing with soap) are not observed following handling of animals, then food, water and milk can be cross-contaminated [56–59]. In our study, many households reported having animals "inside" their household or compound (66%), which could increase the amount of animal faeces near cooking, sleeping, and water/food storage areas. In addition, free-range chickens may also be apt disseminators of *G. intestinalis* cysts throughout the communal environment, other households, and to other animals, as they are often not bound or fenced in [60,61]. Given that pastoralists live in close connection

with their animals and their shared environment, we recommend future studies of IPIs in pastoralist communities consider a One Health approach [9]. Taken together, novel water, sanitation, and hygiene (WASH) methods for semi-mobile (agro-) pastoralist humans and their animals must be considered in order to reduce transmission in these settings.

Children aged 36 to 47 months old had a higher odds of being infected with *A. lumbricoides*. Indeed, a study of children in the Southern Ethiopian region found a peak in infection for *A. lumbricoides* in the 36 to 47 months old age group [16]. However, several other studies have found peaks in younger or in older age groups [17,44], indicating that the specific social and ecological context likely plays a role. As *A. lumbricoides* is transmitted via the faecal-oral route [2,6,9], age-specific changes such as initiation of exploratory behaviour outside of the home, playing with faeces-contaminated soil, and loss of immunologic protection from breastmilk following the weaning period may contribute to a heightened risk in the 36 to 47 month old group [16,44]. In addition, the low rate of exclusive breastfeeding until 6 months (8.4%) and early introduction of animal milks (as early as 1 month) in our study may impact the immune health and nutritional needs of pastoralist children, putting them at greater risk for both IPIs and malnutrition [25,62]. Further qualitative and ethnographic studies are warranted to elucidate the specific factors leading to this result.

There are some limitations in the present study. In this study, severely stunted or severely wasted children were automatically included, while children receiving antibiotics for any type of illness were excluded (which may include symptomatic IPIs). It is therefore possible that we overestimated wasting and stunting and underestimated IPIs in the study population. Further, due to logistical constraints, only one stool sample was able to be analysed per child. Since the diagnostic techniques we used lack perfect sensitivity for some intestinal parasites [33,63], the true prevalence of IPIs may have been underestimated. Given the strong associations of water source, toilet sharing, and cattle and chicken ownership with *G. intestinalis* infection in this study, future studies should include environmental and animal parasitology when assessing parasite prevalence and risk factors in communities in which there is a high rate of contact between humans, animals, and their environment. In addition, the One Health approach in parasitology and pathogenic microorganism research is applicable beyond the few parasites analysed in this study; indeed, there are many parasites and microorganisms of human and animal health importance that are transmitted among and between humans and animals through environmental channels (water, soil, food, plants, air) [6,7,9,40,64,65]. Transmission events between humans, animals, and their environment would best be addressed using genomic techniques [66], and supported with qualitative information to place the transmission events in context and develop appropriate interventions in the given context [67,68]. Taking these approaches would likely improve the health of humans and their animals, as well as reduce the loss of food and income to illness [64,69].

## Conclusions

The prevalence of IPIs in agro-pastoralist and pastoralist children (aged 2–5 years), especially of *G. intestinalis*, are of regional public health concern, given the immediate and long-term health impacts of these types of infections in children. Source of drinking water (rain and river), toilet sharing, and household ownership of cattle and chickens were found to be important risk factors for *G. intestinalis* infection in these communities. Children aged 36 to 47 months were associated with *A. lumbricoides* infection. We recommend additions and improvements to water, sanitation, and hygiene (WASH) infrastructure for use by semi-mobile pastoralists, with attention given to the unique relationship pastoralists have with their animals and environment. Any intervention should be implemented in a transdisciplinary

manner with pastoralists and seek to involve actors from governmental agencies for humans, animals, and the environment, for the improved health of children in pastoralist communities.

## Supporting information

**S1 Fig. Methodological and analytical flow of study.**
(DOCX)

**S2 Fig. Household WASH characteristics of agro-pastoralist and pastoralist children aged 2–5 years living in Adadle woreda, Somali Regional State, Ethiopia.** Alternative informational figure to Table 3, with Somali translations.
(PDF)

**S1 Table. Sociodemographic household characteristics of agro-pastoralist and pastoralist children aged 2–5 years living in Adadle woreda, Somali Regional State, Ethiopia.**
(DOCX)

**S2 Table. Household animal herds of agro-pastoralist and pastoralist children aged 2–5 years of age living in Adadle woreda, Somali Regional State, Ethiopia.**
(DOCX)

**S1 File. Completed STROBE checklist.**
(DOCX)

**S2 File. Manuscript data.** Anonymized study data for reproducing results.
(CSV)

**S3 File. R Master File.** R Markdown master file for reproducing results.
(DOCX)

## Acknowledgments

We would like to thank the agro-pastoralist and pastoralist communities in this study and especially the mothers of the children who allowed their children to participate. Many thanks go to local guides in finding the pastoralist camps, the data collectors, and the laboratory staff at Jigjiga University: Fuad, Ahmed, Dek, Mohammed Hussien, Mohammed Askari, Mohamoud, and countless others. Additional thanks to the Jigjiga One Health Initiative team at Jigjiga University and the Armauer Hansen Research Institute in Ethiopia and the Human and the Animal Health Unit team at Swiss Tropical Public Health Institute in Switzerland.

## Author Contributions

**Conceptualization:** Abdifatah M. Muhummed, Jakob Zinsstag, Jan Hattendorf, Rea Tschopp, Pascale Vonaesch.

**Data curation:** Kayla C. Lanker, Abdifatah M. Muhummed, Jan Hattendorf.

**Formal analysis:** Kayla C. Lanker, Abdifatah M. Muhummed, Jan Hattendorf.

**Funding acquisition:** Jakob Zinsstag, Rea Tschopp, Pascale Vonaesch.

**Investigation:** Abdifatah M. Muhummed, Ramadan Budul Yusuf, Shamil Barsenga Hassen.

**Methodology:** Abdifatah M. Muhummed, Jan Hattendorf, Pascale Vonaesch.

**Project administration:** Abdifatah M. Muhummed, Rea Tschopp, Pascale Vonaesch.

**Supervision:** Abdifatah M. Muhummed, Guéladio Cissé, Jakob Zinsstag, Jan Hattendorf, Rea Tschopp, Pascale Vonaesch.

**Visualization:** Kayla C. Lanker.

**Writing – original draft:** Kayla C. Lanker.

**Writing – review & editing:** Kayla C. Lanker, Abdifatah M. Muhummed, Guéladio Cissé, Jakob Zinsstag, Jan Hattendorf, Ramadan Budul Yusuf, Shamil Barsenga Hassen, Rea Tschopp, Pascale Vonaesch.

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
