## [Decision Letter · Decision Letter 0]

15 Feb 2023

Dear Prof. Vonaesch,

Thank you very much for submitting your manuscript "Prevalence and associated risk factors of intestinal parasitic infections in young pastoralist and agro-pastoralist children in the Somali Region of Ethiopia" for consideration at PLOS Neglected Tropical Diseases. As with all papers reviewed by the journal, your manuscript was reviewed by members of the editorial board and by several independent reviewers. In light of the reviews (below this email), we would like to invite the resubmission of a significantly-revised version that takes into account the reviewers' comments. 

Please kindly attend to the comments raise especially by reviewer #1. Particularly the sample size issue.

We cannot make any decision about publication until we have seen the revised manuscript and your response to the reviewers' comments. Your revised manuscript is also likely to be sent to reviewers for further evaluation.

Sincerely,

Uwem Friday Ekpo, PhD

Academic Editor

Francesca Tamarozzi

Section Editor

Please kindly attend to the comments raise especially by reviewer #1. Particularly the sample size issue.

Reviewer's Responses to Questions

**Key Review Criteria Required for Acceptance?**

**Methods**

-Are the objectives of the study clearly articulated with a clear testable hypothesis stated?

-Is the study design appropriate to address the stated objectives?

-Is the population clearly described and appropriate for the hypothesis being tested?

-Is the sample size sufficient to ensure adequate power to address the hypothesis being tested?

-Were correct statistical analysis used to support conclusions?

-Are there concerns about ethical or regulatory requirements being met?

Reviewer #1: 1. The targeted population is small and restricted to Adadle woreda of the Somali Region. Woreda is the district level and kebele is the smallest administrative unit of Ethiopia. Adadle woreda is one of 10 districts of Shabelle Zone of the Somali Region State of Ethiopia. 

2. Prevalence and risk factors of wasting, stunting and underweight were presented; however, these were not indicated among the objectives of the study.

3. The methods used are appropriate and clearly described.

Reviewer #2: (No Response)

Reviewer #3: The objective of the study is well articulated and match the study design. The population studied is clearly described and is well suited for the study. The sample size is robust and appropriate for the for the statistics carried out

**Results**

-Does the analysis presented match the analysis plan?

-Are the results clearly and completely presented?

-Are the figures (Tables, Images) of sufficient quality for clarity?

Reviewer #1: 1. Table 6: What is the rationale of “Owns cattle” variable grouping and cut-offs? It seems that difference between NO (10%) and YES (14.9%) might be not significant.

2. Line 151: “All screened children who were severely wasted based on their weight for height z-score (WHZ < -2) or severely stunted based on their height for age z-score (HAZ< -2) were automatically included in the study.” How many severely wasted or severely stunted children included? 

3. Lines 248-250: How many severely wasted or severely stunted children included? Because these children were purposively selected, analysis should be adjusted.

4. Table 1: please revise the numbers; e.g. 335 for Complementary food starting; 341 for Complementary milk starting; 338 for vaccination status, etc. It is stated that 345 children had a complete questionnaire and parasitology report (line 223).

5. Table 5: Why some factors are not included in analysis of Ascaria? E.g. Drinking water source and ownership of chicken. All factors should be included regardless significant association. Similarly, some variables in Table 6 are not included in analysis of Giardia; e.g. shared toilet and household owns soap. 

6. Tables 5 and 6: add columns for P values for bivariate and multivariate. Although it can be judged from the 95%CI, but it still useful.

Reviewer #2: (No Response)

Reviewer #3: The results are well presented with appropriate figures and tables

**Conclusions**

-Are the conclusions supported by the data presented?

-Are the limitations of analysis clearly described?

-Do the authors discuss how these data can be helpful to advance our understanding of the topic under study?

-Is public health relevance addressed?

Reviewer #1: 1. Lines 44-45: “employing a One Health approach to treat children and animals for parasitic infections are important” this is not supported by the findings. Despite of significant associations detected, the study design does not allow causal inference. This is also applied to discussion section. This is also applied for main conclusion section.

2. Discussion about the association found between owning chicken and Giardia infection lacks in depth explanation. 

3. Lines 454-467: this paragraph about nutritional status can be removed. It is recommended to add related variables to analysis on stunting and wasting and then discuss the association between IPIs and nutritional status. Table 1 S shows negative association between Giardia and Ascaris with wasting; wasting can be added as a variable in Tables 5 and 6. Malnutrition and IPIs are coexisted problems and one is causative for the other.

Reviewer #2: (No Response)

Reviewer #3: The conclusion in this study is supported by the data presented and limitations of the study stated. The public health importance is clearly emphasized

**Editorial and Data Presentation Modifications?**

Reviewer #1: 1. The title should be rephrased. The subjects and targeted populations are unclear. It should be “…… among young children in pastoralist and agro-pastoralist populations (or can be communities) in the Somali Region of Ethiopia”. This is also applied for related sentences in the text.

2. Giardia lamblia: lamblia is an old name; you may use Giardia intestinalis throughout the manuscript.

3. Line 34: state the overall prevalence first; i.e. those infected with at least one parasite species.

4. Lines 37-43: sentences about risk factors can be rephrased. Just indicate the risk factors with respective statistics (aOR & 95%CI); no need to mention the counterpart groups.

5. Line 78: “Most human intestinal parasites require an intermediate animal host to complete their lifecycle,” this is not accurate; at least for the parasites detected in this study, even Hymenolepis nana can be directly transmitted.

6. Lines 76-82: this paragraph can be removed, unless information on infections among animals can be added.

7. Line 321: state the overall prevalence first; i.e. those infected with at least one parasite species.

8. Tables 5 and 6: modify the title to include the bivariate and multivariate analyses.

9. Table 5: on positive case is missing. The total is 79 while it is 80 in Table 4.

10. Tables 5 and 6: remove the % sign from the results of 2nd column.

11. Lines 380-384: avoid repeating results; focus on main findings only.

Reviewer #2: (No Response)

Reviewer #3: Minor revision

**Summary and General Comments**

Reviewer #1: 1- The study and findings are not novel.

2- Anthropometric indices were calculated and results on the prevalence and associated factors of wasting, stunting and underweight were presented and discussed with few related supplementary tables were provided. However, these were not indicated among the objectives of the study. However, stunting, wasting and underweight should be added as variables or potential predictors of Giardia and Ascaris (tables 5 & 6). Otherwise, text and results about nutritional status should be removed from the manuscript.

3- Rationale of “Owns cattle” grouping and cut-off are unclear. 

4- This is a small regional study among small population restricted to Adadle woreda of the Somali Region. Woreda is the district level and kebele is the smallest administrative unit of Ethiopia. Adadle woreda is one of 10 districts of Shabelle Zone of the Somali Region State of Ethiopia.

Reviewer #2: (No Response)

Reviewer #3: The MS titled: Prevalence and associated risk factors of intestinal parasitic infections in young pastoralist and agro-pastoralist children in the Somali Region of Ethiopia is well written and the finding in this study might be useful to the parasitologists, public health practitioners and public in general. Though there are observation and question as highlighted in the MS. The observations are as follow

1. Line 25: Consider deleting young from the young children

2. Line 67: As above

3. Line 78: separate the lifecycle

4. Line 102: Consider: The study assessed the prevalence of intestinal.........

5.Line 115: Complete the: obtained from the parent

6. Line 126: Mention the annual rainfall too

7. Line 193: Delete the highlighted and start with: The samples.......

8. Line 323: Change infestation to infection

9. Line 326: How did you identify to species level the Entamoeba? State this in your material and method. But if the method used could only identify to genus level then just refer to Entamoeba spp. in your result and subsequent discussion

10. Line 330: You referred to detection of the helminth parasites. Was it the adult parasites you detected or their eggs? if it is their eggs, then refer to the eggs and not the parasite. Also, the word hookworms is wide, mention the hookworm eggs detected. In that section also, you did not define H. nana

Line 372-376: The first two statements contradict, hence stick with one

Line 380-384: The highlighted are results, delete. Also, you dwell so much on result than the discussion in discussion section

Line 459: due seasonal , should be changed to, due to seasonal

PLOS authors have the option to publish the peer review history of their article (what does this mean?). If published, this will include your full peer review and any attached files.

Reviewer #1: No

Reviewer #2: Yes: Endalkachew Nibret

Reviewer #3: Yes: Takeet Michael Irewole
---

## [Decision Letter · Decision Letter 1]

15 May 2023

Dear Prof. Vonaesch,

Thank you very much for submitting your manuscript "Prevalence and associated risk factors of intestinal parasitic infections among children in pastoralist and agro-pastoralist communities in the Adadle woreda of the Somali Region of Ethiopia" for consideration at PLOS Neglected Tropical Diseases. As with all papers reviewed by the journal, your manuscript was reviewed by members of the editorial board and by several independent reviewers. The reviewers appreciated the attention to an important topic. Based on the reviews, we are likely to accept this manuscript for publication, providing that you modify the manuscript according to the review recommendations. 

Sincerely,

Uwem Friday Ekpo, PhD

Academic Editor

Francesca Tamarozzi

Section Editor

Reviewer's Responses to Questions

**Key Review Criteria Required for Acceptance?**

**Methods**

-Are the objectives of the study clearly articulated with a clear testable hypothesis stated?

-Is the study design appropriate to address the stated objectives?

-Is the population clearly described and appropriate for the hypothesis being tested?

-Is the sample size sufficient to ensure adequate power to address the hypothesis being tested?

-Were correct statistical analysis used to support conclusions?

-Are there concerns about ethical or regulatory requirements being met?

Reviewer #1: None

Reviewer #2: (No Response)

Reviewer #3: The objectives are clearly stated andthe study designed appropriately addressed the objectives. The population is clearly defined but the expected prevalence used to calculate the sample size is inappropriate. The sample size is adequate and no concern about the ethical requirement

**Results**

-Does the analysis presented match the analysis plan?

-Are the results clearly and completely presented?

-Are the figures (Tables, Images) of sufficient quality for clarity?

Reviewer #1: None

Reviewer #2: (No Response)

Reviewer #3: The result is completely presented and analysed with appropriate tool. The figures and tables are adequate and clearly presented

**Conclusions**

-Are the conclusions supported by the data presented?

-Are the limitations of analysis clearly described?

-Do the authors discuss how these data can be helpful to advance our understanding of the topic under study?

-Is public health relevance addressed?

Reviewer #1: None

Reviewer #2: (No Response)

Reviewer #3: The conclusion support the data presented and limitation well spelt out

**Editorial and Data Presentation Modifications?**

Reviewer #1: None

Reviewer #2: (No Response)

Reviewer #3: (No Response)

**Summary and General Comments**

Reviewer #1: In the current revised manuscript, the authors have addressed my comments and I am satisfied with the way they have improved the manuscript. However, few minor corrections need to be considered.

1- Lines 45-48: Conclusions’ statement can be further modified to “Improving access to safe water, sanitation, and hygiene services in Adadle and employing a One Health approach would likely improve the health of children living in (agro-) pastoralist communities in Adadle and ESRS; however, further studies are required.”.

2- Line 460: “…. should be considered ….”

3- Line 462: “…….. were more likely to be infected with ….”. Also, modify the sentence in line 502.

4- Line 475: “There are some limitations in the present study ….”

5- Line 475: remove “First”.

6- Lines 476-479: this long sentence can be split in order to improve clarity.

7- Line 500: “Source of drinking water (rain and river), …”

8- IPI or IPIs? Please ensure consistency throughout the entire text.

9- Lines 39, 203, 341, 342, and Table 4: change to “hookworm (Ancylostoma spp. / Necator americanus)”

10- Line 347: “……; hookworm 2/16.”

Reviewer #2: (No Response)

Reviewer #3: General comment: The MS is well structured but full of grammatical errors and expressions that need to be corrected and as such the authors may need native English expert. Also, the materials and methods is not explicit especially in the area of parasite identification to the species level. 

My specific comments areas follow:

a. Line 25: Delete young from the young children

b. Line 34: Re-cast as: Protozoan parasite prevalence was 24.9% and 21.9% for E. hystolitica/dispar and Giardia lambda, respectively.

c. Line 39-41: The highlighted should be recast for clarity

d. Line 50: Delete, in

e. Line 53: Delete the highlighted

f. Line 67: Consider deleting the: Young

g. Line 115-116: The highlighted should read: obtained from theparent

h. Line122-128: Include the map of the study area

i. Line 133-134: You cannot use expected prevalence to calculate the sample size but already established prevalence in that area or elsewhere

j. Line 156-157:If you want to refer to young children, then you must define the age group that constitute the young and the non-young children

k. Line 194: You may consider deleting : and aliquoted

l. Line 202: Do you mean : Normal saline?

m. Line 204: At what magnification

n. Line 206: Delete: stool

o. Line 207: Name the antiparasitic agent used

p. Line 222: Delete: (N=366)

q. Line 233: Delete; even

r. Line 321: First, indicate the overall prevalence (Single+multiple infection) before stratifying to single, multiple prevalences

s. Line 325: How did you differentiate to species level? Because you used microscopy

t. Generally, in the result section, I think the detection of the Giardia and Entameba should stop at the genus level. Also, pictures of the detected parasites would add a lot of value to this MS. I also think that you should mention the hookworms encountered in this study instead of lobbing all of them together.

u. Line 372-378: The highlighted should come towards the end before concluding the discussion 

v. Line 380-384:The highlighted are results that you have mentioned in the result section. So, consider deleting and just discuss what could be responsible for the prevalence reported in this study compared to what has been done elsewhere

PLOS authors have the option to publish the peer review history of their article (what does this mean?). If published, this will include your full peer review and any attached files.

Reviewer #1: Yes: Hesham M. Al-Mekhlafi

Reviewer #2: Yes: Endalkachew Nibret

Reviewer #3: Yes: Takeet, Michael Irewole

Figure Files:

Data Requirements:

Reproducibility:

References

---

## [Decision Letter · Decision Letter 2]

8 Jun 2023

Dear Prof. Vonaesch,

We are pleased to inform you that your manuscript 'Prevalence and associated risk factors of intestinal parasitic infections among children in pastoralist and agro-pastoralist communities in the Adadle woreda of the Somali Region of Ethiopia' has been provisionally accepted for publication in PLOS Neglected Tropical Diseases.

Best regards,

Uwem Friday Ekpo, PhD

Academic Editor

Francesca Tamarozzi

Section Editor

Reviewer's Responses to Questions

**Key Review Criteria Required for Acceptance?**

**Methods**

-Are the objectives of the study clearly articulated with a clear testable hypothesis stated?

-Is the study design appropriate to address the stated objectives?

-Is the population clearly described and appropriate for the hypothesis being tested?

-Is the sample size sufficient to ensure adequate power to address the hypothesis being tested?

-Were correct statistical analysis used to support conclusions?

-Are there concerns about ethical or regulatory requirements being met?

Reviewer #1: None

**Results**

-Does the analysis presented match the analysis plan?

-Are the results clearly and completely presented?

-Are the figures (Tables, Images) of sufficient quality for clarity?

Reviewer #1: None

**Conclusions**

-Are the conclusions supported by the data presented?

-Are the limitations of analysis clearly described?

-Do the authors discuss how these data can be helpful to advance our understanding of the topic under study?

-Is public health relevance addressed?

Reviewer #1: None

**Editorial and Data Presentation Modifications?**

Reviewer #1: None

**Summary and General Comments**

Reviewer #1: All my comments have been considered.

PLOS authors have the option to publish the peer review history of their article (what does this mean?). If published, this will include your full peer review and any attached files.

Reviewer #1: **Yes: **Hesham M. Al-Mekhlafi

---

## [Editor Report · Acceptance letter]

28 Jun 2023

Dear Prof. Vonaesch,

We are delighted to inform you that your manuscript, "Prevalence and associated risk factors of intestinal parasitic infections among children in pastoralist and agro-pastoralist communities in the Adadle woreda of the Somali Regional State of Ethiopia," has been formally accepted for publication in PLOS Neglected Tropical Diseases.

Best regards,

Shaden Kamhawi

co-Editor-in-Chief

Paul Brindley

co-Editor-in-Chief
